# Correlation between Remote Dielectric Sensing and Chest X-Ray to Assess Pulmonary Congestion

**DOI:** 10.3390/jcm12020598

**Published:** 2023-01-11

**Authors:** Toshihide Izumida, Teruhiko Imamura, Masakazu Hori, Masaki Nakagaito, Hiroshi Onoda, Shuhei Tanaka, Ryuichi Ushijima, Koichiro Kinugawa

**Affiliations:** 1Second Department of Internal Medicine, University of Toyama, 2630 Sugitani Toyama, Toyama 930-0194, Japan; 2Department of Cardiology, Japanese Red Cross Takayama Hospital, 3-113-11 Tenman-machi Takayama, Gifu 506-8550, Japan

**Keywords:** cardiology, pulmonary edema, heart failure, monitoring

## Abstract

Background: Chest X-ray is a practical tool to semi-qualify pulmonary congestion. Remote dielectric sensing (ReDS) is a recently introduced, non-invasive, electromagnetic energy-based technology to quantify pulmonary congestion without expert technique. We compared these two modalities to clarify appropriate clinical situations for each modality. Methods: ReDS and chest X-ray measurements were prospectively performed on admission in consecutive hospitalized patients with cardiovascular diseases. In the chest X-ray, the congestive score index (CSI) was calculated blindly by two independent experts and averaged. CSIs were correlated with ReDS values. Results: A total of 458 patients (76 (69, 82) years old, 267 men, and 130 heart failure) were included. Median ReDS value was 28% (25%, 33%). There was a mild correlation between ReDS values and CSIs (r = 0.329, *p* < 0.001). The correlation between ReDS values and CSIs became stronger in the heart failure cohort (r = 0.538, *p* < 0.001). In patients with mild congestion (ReDS < 35%), ReDS values, instead of CSI, stratified the degree of congestion. In patients with severe congestion (ReDS > 35%), both modalities stratified the degree of congestion. Conclusions: Both chest X-ray and ReDS are useful for assessing severe pulmonary congestion, whereas ReDS would be preferred to chest X-ray in stratifying the severity of mild pulmonary congestion.

## 1. Introduction

Chest X-ray (CXR) is a practical and established tool to assess pulmonary congestion during daily clinical practice [1]. The existence of severe congestion is obvious in CXR, displayed as a butterfly shadow, whereas the detailed assessment of mild congestion is challenging and requires expert technique [1,2].

Recently, congestion score index (CSI), which is a semi-quantitative approach to evaluate pulmonary congestion based on a 6-segment evaluation of CXR, scoring each zone from 0 (no congestion) to 3 (intense alveolar pulmonary edema), has been proposed to overcome the above challenges. Several studies have demonstrated its utility for assessing pulmonary congestion [3,4,5].

Remote dielectric sensing (ReDS) has been established as a noninvasive methodology to quantify the degree of pulmonary congestion without expert technique [6]. Its utility was validated by comparing it with other modalities, including high-resolution computed tomography and invasive hemodynamics assessment [7,8]. Thus, the ReDS system has been established as a noninvasive tool to quantify lung fluid amounts. Given its high sensitivity, ReDS might particularly be applicable to those with mild congestion. The device has been commercially available in Japan since 2022, at approximately $40,000 per device.

However, the clinical utility of these two modalities in each clinical scenario remains uncertain: i.e., none, mild, or severe pulmonary congestion with/without heart failure [HF]). We aimed to correlate ReDS and CSI in patients with and without HF and discuss the appropriate clinical situations for each modality.

## 2. Materials and Methods

### 2.1. Patient Selection

From August 2021 to July 2022, consecutive patients who were hospitalized in our institute to examine and treat cardiovascular diseases were included in our prospective study. Successive measurements of ReDS values and CXR were performed on admission in all participants. Patients who could not wear ReDS vests or had obvious lung diseases, such as acute pneumoniae, lung cancer, interstitial lung disease, asthma, and chronic obstructive pulmonary disease, were excluded. HF was defined according to the Framingham’s criteria. All patients provided written informed consent before inclusion. The study protocol complied with the Declaration of Helsinki and was approved by the Ethic Committee of the University of Toyama (MTK2020007).

### 2.2. Measurement of ReDS

ReDS is a noninvasive system to evaluate lung fluid content by analyzing the dielectric properties of the lung portion using sonar-released, low-power electromagnetic signals. The ReDS value was taken by wearing the vest and maintaining a resting and sitting position for one minute (Figure 1). The manufacturer proposes that the normal range for ReDS values is between 20% and 35%. The definition of significant pulmonary congestion is an ReDS value of 35% or higher.

### 2.3. Measurement of Congestion Score Index of Chest X-Ray

The severity of the radiographic pulmonary congestion was evaluated using a previously reported CSI (Figure 2) [3,4,5]. The scoring of the congestion was performed on each 6-segment of lung fields. The grades of congestion were defined as follows; 0 = normal, 1 = cephalization, perihilar haze, peribronchial cuffing, or Kerley lines; 2 = interstitial pulmonary edema or localized/confluent mild edema; 3 = confluent intense edema. The total score ranging from 0 to 18 was obtained. If a segment was covered with pleural effusion, the segment was not scored. The CSI was calculated as the total congestion score divided by number of available lung segments. Two independent, experienced physicians (TIz and TIm) blinded to the clinical parameters evaluated the CSI of all patients, and the obtained CSIs were averaged.

### 2.4. Statistical Analysis

Continuous variables were presented as median and interquartile, and categorical variables were presented as numbers and percentages. Pearson’s correlation coefficient was used to evaluate correlations between ReDS values and CSIs. The correlations were reevaluated among HF patients. Statistical analyses were performed using SPSS Statistics 26 (IBM, Armonk, NY, USA). Two-sided *p*-values < 0.05 were considered statistically significant.

## 3. Results

### 3.1. Baseline Characteristics

A total of 500 consecutive patients were considered to be included in this prospective study. Of them, 42 patients were excluded due to inappropriate physical health or lung diseases. Ultimately, 458 patients were included (76 [69, 82] years old, 267 men [58%]) (Table 1). The median body mass index (BMI) was 22.8 (20.5, 25.2) kg/m^2^. The median ReDS value was 28% (25%, 33%), and the median CSI was 0.08 (0.00, 0.25). Our study included 130 HF patients (28%). The κ value to assess inter-observer reliability between the two readers was 0.88 (95% confidence interval, 0.86–0.90) for the scoring of CSI.

### 3.2. ReDS Values and CSI among Those without Lung Diseases

There was a mild correlation between ReDS values and CSI among all 458 patients (r = 0.329, *p* < 0.001; Figure 3A). In patients with normal ReDS ranges (i.e., between 20% and 35%), CSI remained relatively flat irrespective of any ReDS values (r = −0.007, *p* = 0.891, *n* = 391). In patients with significant congestion (i.e., ReDS > 35%), ReDS had a strong linear correlation with CSI (r = 0.593, *p* < 0.001, *n* = 67; Table A1). The correlation between ReDS values and CSI became stronger in the HF cohort (r = 0.538, *p* < 0.001) (Figure 3B). In HF patients with significant congestion (i.e., ReDS > 35%), ReDS had a further stronger linear correlation with CSI (r = 0.705, *p* < 0.001, *n* = 23).

## 4. Discussion

We investigated the correlations between ReDS values, which was measured by novel ReDS technology, and CSI, which was measured by experienced cardiologists from CXR, in the assessment of pulmonary congestion. There was a mild correlation between ReDS values and CSIs in all participants. In patients with normal ReDS values (≤35% ReDS values), CSI remained at relatively low levels irrespective of a variety of ReDS values. In patients with significant pulmonary congestion with ReDS > 35%, ReDS values had a linear and strong correlation with CSI.

### 4.1. ReDS Values and CSIs to Evaluate Pulmonary Congestion

The accurate assessment of pulmonary congestion is a key to risk stratification and constructing tailor-made therapeutic strategies in patients with congestive HF [2,9]. However, there is no established gold standard to quantify the degree of pulmonary congestion thus far.

ReDS technology is a recently introduced, novel modality to quantify the percentage of lung fluid noninvasively and without expert technique. ReDS values have a moderate correlation with high-resolution computed tomography, a mild correlation with invasively measured pulmonary artery wedge pressure, and weak correlations with plasma B-type natriuretic peptide levels and lung ultrasound [10,11,12]. Consistently, patients with ReDS values > 35% did not necessarily have extremely elevated plasma B-type natriuretic peptide levels in this study. CXR is one of the most practical tools to assess pulmonary congestion in daily clinical practice. However, no studies have assessed the concordance between ReDS technology and CXR interpretation. This is a rationale of this study.

The correlation between ReDS values and CSIs was mild. Of note, CSI remained at low levels when pulmonary congestion was nonexistent or mild (i.e., ReDS values ≤ 35%). Nevertheless, ReDS values presented a variety of levels. In other words, ReDS system may stratify even mild pulmonary congestion, which CXR cannot distinguish. This may not be surprising, because ReDS can detect only a slight degree of pulmonary congestion. On the other hand, it would be challenging to distinguish mild pulmonary congestion from the physiological consolidation of lung vasculatures in CXR.

In patients with significant pulmonary congestion (i.e., ReDS values > 35%), ReDS values and CSIs showed a stronger linear correlation. As experienced in daily clinical practice, the apparent and broad lung shadow can be easily diagnosed with severe pulmonary congestion in CXR. 

### 4.2. Clinical Implication of ReDS Measurement

We do not deny the usefulness of CXR in daily practice to assess pulmonary congestion or any other cardio-pulmonary abnormalities [2]. Of note, we excluded those with any lung diseases, which can be assessed in detail by CXR. CXR is relatively easy to perform and the exposure to radiation is minimum. However, one of the critical drawbacks of this modality should be its subjectivity depending on who read the CXR. In this study, we obtained satisfactory inter-observer reliability, given that both reviewers were well-experienced cardiologists. Similar reliability may not be achieved by other non-expert readers.

ReDS values showed good agreement with CSIs which were scored by expert cardiologists, particularly among those with severe pulmonary congestion and/or HF. ReDS may be a good alternative to CXR, particularly when repeated CXR is hesitated or unavailable in this cohort.

Additionally, ReDS technology can stratify those with no or mild pulmonary congestion. Such mild pulmonary congestion may also be caused by non-HF etiologies, including chronic kidney diseases. ReDS technology may have advantage over CXR in accurately quantifying the degree of mild pulmonary congestion and adjusting the doses of diuretics. Of note, only a slight residual pulmonary congestion at the index discharge can worsen clinical outcomes [9]. ReDS technology, rather than CXR, is particularly useful to quantify residual pulmonary congestion prior to the index discharge. Nevertheless, there are no studies that directly investigated the clinical implication of mild pulmonary congestion assessed by ReDS.

### 4.3. Study Limitations

There are several concerns and limitations. Since our study was a single center study, we require further studies to evaluate the scientific rigor or external validity. We included only in-hospital patients. The applicability of our findings in remote monitoring and home care settings is the next concern [13]. We excluded those with lung diseases, given that ReDS technology is not an imaging modality. Thus, other imaging modalities, including CXR and lung ultrasound, should be concomitantly applied for the assessment of a variety of cardio-pulmonary diseases. We did not routinely obtain physical examinations to assess pulmonary congestion, although it is sometimes challenging to accurately estimate the degree of pulmonary congestion by physical examinations alone. 

## 5. Conclusions

Both CXR and ReDS are useful to assess severe pulmonary congestion, whereas ReDS is preferred to CXR in stratifying the severity of mild pulmonary congestion. 

## Figures and Tables

**Figure 1 jcm-12-00598-f001:**
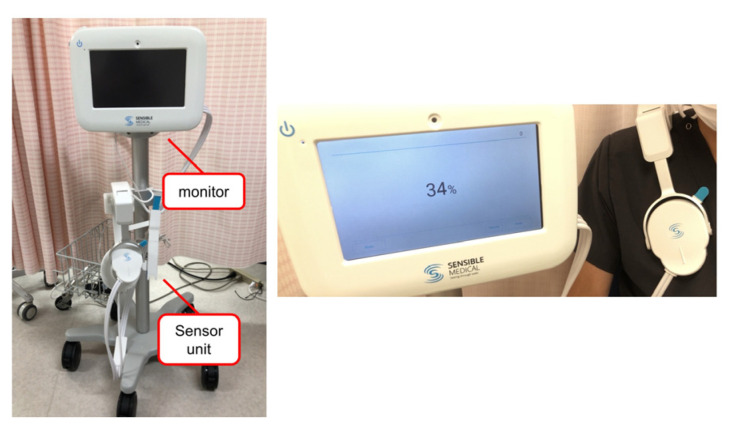
How to measure remoting dielectric sensing (ReDS) value. The device consists of a sensor unit and a monitor to display the results. Patients wear the sensor unit and wait for approximately one minute. ReDS value is displayed immediately after the measurement.

**Figure 2 jcm-12-00598-f002:**
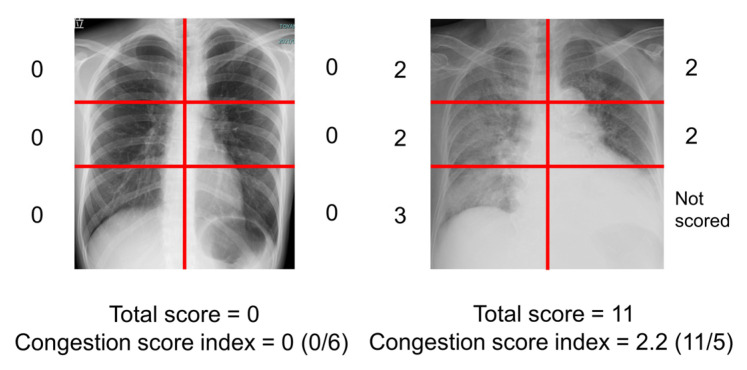
How to measure congestion score index of chest X-ray (examples are displayed). The scoring of congestion was performed on each 6-segment of lung fields and the grades of congestion were defined as follows; 0 = normal, 1 = cephalization, perihilar haze, peribronchial cuffing, or Kerley lines; 2 = interstitial pulmonary edema and localized or confluent mild edema; 3 = confluent intense edema. The total score was obtained. If a segment covered with pleural effusion, the segment was not scored. The congestion score index was calculated as the total score divided by number of available lung segments.

**Figure 3 jcm-12-00598-f003:**
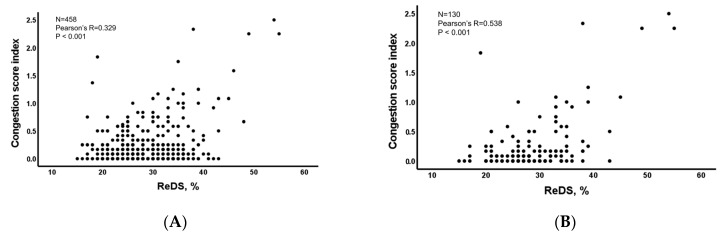
(**A**) The correlation between ReDS values and congestion score index in patients without lung diseases, (**B**) the correlation between ReDS values and congestion score index in heart failure patients without lung diseases. *p*-value < 0.05 by Pearson’s correlation.

**Table 1 jcm-12-00598-t001:** Baseline characteristics.

	All (*n* = 458)	Non-Heart Failure (*n* = 328)	Heart Failure (*n* = 130)
Demographics			
Age, years	76 (69, 82)	76 (69, 82)	78 (64, 83)
Man	267 (58%)	180 (55%)	87 (67%)
Height, cm	159 (151, 166)	159 (151, 166)	158 (152, 169)
Body mass index, kg/m^2^	22.8 (20.5, 25.2)	23.0 (20.9, 25.3)	21.9 (19.4, 25.0)
Laboratory data			
Hemoglobin, g/dL	12.5 (11.1, 13.8)	12.5 (11.1, 13.7)	12.7 (11.3, 14.0)
Serum albumin, g/dL	3.9 (3.6, 4.2)	3.9 (3.6, 4.2)	3.8 (3.5, 4.1)
Serum creatinine, mg/dL	1.0 (0.8, 1.5)	0.9 (0.8, 1.4)	1.2 (0.9, 1.6)
Plasma B-type natriuretic peptide, pg/mL	95 (33, 253)	69 (25, 156)	285 (100, 604)
Echocardiographic data			
Left ventricular ejection fraction, %	62 (51, 69)	66 (59, 71)	43 (32, 49)
Left ventricular end-diastolic diameter, mm	48 (43, 53)	46 (42, 50)	52 (48, 61)
Left ventricular end-systolic diameter, mm	31 (27, 38)	29 (26, 33)	42 (35, 50)
Left atrial diameter, mm	41 (35, 47)	40 (34, 46)	43 (37, 49)
Past medical history			
Heart failure	130 (28%)		
Stroke	84 (18%)	67 (20%)	17 (13%)
History of coronary intervention	107 (23%)	77 (23%)	30 (23%)
Hypertension	339 (74%)	247 (75%)	92 (71%)
Dyslipidemia	251 (55%)	182 (55%)	69 (53%)
Diabetes mellitus	163 (36%)	111 (34%)	52 (40%)
Valvular diseases	149 (33%)	90 (27%)	59 (45%)
Chronic kidney diseases	305 (66%)	200 (61%)	105 (81%)
Atrial fibrillation	161 (35%)	102 (31%)	59 (45%)
ReDS, %	28 (25, 33)	28 (24, 32)	28 (25, 33)
Chest X-ray			
Congestion score index	0.08 (0.00, 0.25)	0.00 (0.00, 0.17)	0.08 (0.00, 0.27)

Continuous variables were expressed as median and interquartile, and categorical variables were expressed as numbers and percentages. Chronic kidney disease was defined as estimated glomerular filtration rate less than 60 mL/min per 1.73 m^2^.

## Data Availability

Data are available upon appropriate request.

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
