# Peer review of "Correlation between Remote Dielectric Sensing and Chest X-Ray to Assess Pulmonary Congestion"

_jcm, 2023, doi:10.3390/jcm12020598_

Round 1
Reviewer 1 Report (New Reviewer)
Overall a very interesting paper with potential value of the ReDS especially in the outpatient setting and before discharge.
1. Can you please provide information in your article about the cost of the device/ unit?
2. In paragraph 4.2 author mentioned reliability and non-expert readers. This is a strong statement. Most of the CXR readers are chest radiologists. It is better if the author rephrase it to depict the subjectivity depending who is reading the CXR rather than state about non-expert readers.
3. Lines 185-186 spell check "... this is the large study of ReDS in 500 cases, and it would be a useful study...", Line 196 spell check "... expart technique..."
Author Response
Reviewer1
General comment
Overall a very interesting paper with potential value of the ReDS especially in the outpatient setting and before discharge.
General response
We sincerely express our great appreciation for the critical comments. We attempted our best to respond to the comments and revise our draft. Please read our responses and updated draft.
Comment 1
Can you please provide information in your article about the cost of the device/ unit?
Response 1
Thank you for the comment. The device is commercially available in Japan from 2022 by approximately $40,000 per device.
Before 1
- Introduction
None.
After 1
- Introduction
The device is commercially available in Japan from 2022 by approximately $40,000 per device.
Comment 2
In paragraph 4.2 author mentioned reliability and non-expert readers. This is a strong statement. Most of the CXR readers are chest radiologists. It is better if the author rephrase it to depict the subjectivity depending who is reading the CXR rather than state about non-expert readers.
Response 2
We appreciate the comment. It might vary in each country and institute who read CXR. For example, in Japan, CXR is NOT read by radiologists. Anyway, we agree with the reviewer that the phrase is so strong. One of the issues of CXR would be subjectivity depending on who read CXR. We rephrased the sentence.
Before 2
4.2 Clinical implication of ReDS measurement:
However, one of the critical drawbacks of this modality should be a requirement of expert readers.
After 2
4.2 Clinical implication of ReDS measurement:
However, one of the critical drawbacks of this modality should be its subjectivity depending on who read CXR.
Comment 3
Lines 185-186 spell check "... this is the large study of ReDS in 500 cases, and it would be a useful study...", Line 196 spell check "... expart technique...
Response 3
We appreciate the comment. We corrected typographs and grammatical errors. We believe that the revised draft would be suitable for the publication.
Before 3
4.3 Study limitations:
However, this is the large study of ReDS in 500 cases, and it would be a useful study that might demonstrate the clinical significance of its usefulness in remote monitoring and home-care settings for managing pulmonary congestion.[11]
- Conclusions
The ReDS technology might be a practical alternative tool to non-invasively quantify the wide degree of pulmonary congestion without expart technique, as compared to conventional CXR.
After 3
4.3 Study limitations:
We included only in-hospital patients. Applicability of our findings in remote monitoring and home-care settings remains the next concern.[13]
- Conclusions
Both CXR and ReDS are useful to assess severe pulmonary congestion, whereas ReDS is preferred to CXR in stratifying the severity of mild pulmonary congestion.

Reviewer 2 Report (New Reviewer)
Correlation between Remote Dielectric Sensing and Chest X-Ray to Assess Pulmonary Congestion
Dr Izumida T et al., report a prospective cohort study assessing the correlation between remote dielectric sensing (ReDS) and chest X-ray-derived congestion score index (CSI) in patients hospitalized for cardiovascular diseases. Their results showed significant correlation between these congestion markers (r=0.33, p<0.001), and in heart failure (HF) (r=0.54, p<0.001). The overall manuscript reads well. Nonetheless, I have major comments and remarks.
MAJOR COMMENTS:
1/ If patients have mild pulmonary congestion as assessed by ReDS, I am not completely convinced what is clinical implications of ReDS-derived pulmonary congestion. Do ReDS may stratify patient risk of developing or recurrent HF hospitalization even if ReDS is low? Otherwise, their conclusion should be modified; “Both chest X-ray and ReDS would be useful to assess severe pulmonary congestion, whereas ReDS would be preferred to chest X-ray in stratifying the severity of mild pulmonary congestion.”
2/ Please show correlation between ReDS values, CSI and natriuretic peptide. Although natriuretic peptide may be confounded by several clinical presentations (i.e., old age, obesity and renal function), this biomarker has established diagnostic and prognostic values in HF.
3/ In this study, congestion was defined by a ReDS value (≥35%). Although patients had significant pulmonary congestion as assessed by ReDS, median BNP value of such patients was relatively low (124pg/ml). Please comment on this issue, comparing prior studies for patients hospitalized for worsening HF.
4/ Recently published data showed the usefulness of lung ultrasound to detect slightly developed pulmonary congestion with high sensitivity and specificity in HF. Please discuss this point.
5/ Clinical signs and symptoms of congestion (i.e., leg edema, jugular venous distention, and New York Heart Association class) were lacking in the current analysis.
6/ >70% of patients were not diagnosed with HF, but had certain degrees of ReDS-derived pulmonary congestion as presented in Table 1. So, what is a diagnosis for such patients? In general, patients who have symptomatic congestion are defined as HF.
7/ Renal function may influence the relationship between ReDS and CSI particularly in patients with HF.
MINOR COMMENTS:
1/ Please check prevalence rate of prior heart failure in Table 1. It may be typo error.
Author Response
Reviewer 2
General comment
Dr Izumida T et al., report a prospective cohort study assessing the correlation between remote dielectric sensing (ReDS) and chest X-ray-derived congestion score index (CSI) in patients hospitalized for cardiovascular diseases. Their results showed significant correlation between these congestion markers (r=0.33, p<0.001), and in heart failure (HF) (r=0.54, p<0.001). The overall manuscript reads well. Nonetheless, I have major comments and remarks.
General response
We sincerely express our great appreciation for the critical comments. We attempted our best to respond to the comments and revise our draft. We also corrected typographs and grammatical errors. We believe that the revised draft would be suitable for the publication. Please read our responses and updated draft.
Comment 1
If patients have mild pulmonary congestion as assessed by ReDS, I am not completely convinced what is clinical implications of ReDS-derived pulmonary congestion. Do ReDS may stratify patient risk of developing or recurrent HF hospitalization even if ReDS is low? Otherwise, their conclusion should be modified; “Both chest X-ray and ReDS would be useful to assess severe pulmonary congestion, whereas ReDS would be preferred to chest X-ray in stratifying the severity of mild pulmonary congestion.”
Response 1
We appreciate the comment. We completely agree with the reviewer. There are few studies that investigated the prognostic impact of mild pulmonary congestion indicated by ReDS values below 35%. We revised the conclusion section as suggested.
Before 1
- Conclusions
The ReDS technology might be a practical alternative tool to non-invasively quantify the wide degree of pulmonary congestion without expert technique, as compared to conventional CXR.
After 1
- Conclusions
Both CXR and ReDS would be useful to assess severe pulmonary congestion, whereas ReDS would be preferred to CXR in stratifying the severity of mild pulmonary congestion.
Comment 2
Please show correlation between ReDS values, CSI and natriuretic peptide. Although natriuretic peptide may be confounded by several clinical presentations (i.e., old age, obesity and renal function), this biomarker has established diagnostic and prognostic values in HF.
Response 2
We appreciate your important comment. We previously reported a study evaluating the correlation between ReDS values and BNP levels (Int Heart J 2022, PMID:36450552). In this study, ReDS values and BNP levels had only a weak correlation. BNP level is dominantly affected by the intra-cardiac “pressure”, whereas ReDS values indicate the “volume” of pulmonary fluid. Furthermore, as suggested by the reviewer, BNP can be confounded by several clinical parameters including age, obesity, the existence of atrial fibrillation, and renal impairment. We revised the discussion section to mention this previous finding.
Before 2
4.1 ReDS values and CSIs to evaluate pulmonary congestion:
ReDS values have a moderate correlation with high-resolution computed tomography and a mild correlation with invasively measured pulmonary artery wedge pressure.
After 2
4.1 ReDS values and CSIs to evaluate pulmonary congestion:
ReDS values have a moderate correlation with high-resolution computed tomography, a mild correlation with invasively measured pulmonary artery wedge pressure, and weak correlations with plasma B-type natriuretic peptide levels and lung ultrasound [10-12].
Comment 3
In this study, congestion was defined by a ReDS value (≥35%). Although patients had significant pulmonary congestion as assessed by ReDS, median BNP value of such patients was relatively low (124pg/ml). Please comment on this issue, comparing prior studies for patients hospitalized for worsening HF.
Response 3
We appreciate your comment. As we responded above, there is only a weak correlation between ReDS values and BNP levels (Int Heart J 2022, PMID:36450552). Again, ReDS values indicate pulmonary fluid “volume” and BNP levels represent intra-cardiac “pressure”. Please see a below figure retrieved from this previous paper. Plasma BNP levels distribute widely in each ReDS value.
Thus, it might not be so surprising that those with ReDS values >35% in this study had relatively lower BNP levels.
Before 3
None.
After 3
4.2 Clinical implication of ReDS measurement:
Consistently, patients with ReDS values >35% did not necessarily have extremely elevated plasma B-type natriuretic peptide levels in this study.
Comment 4
Recently published data showed the usefulness of lung ultrasound to detect slightly developed pulmonary congestion with high sensitivity and specificity in HF. Please discuss this point.
Response 4
We appreciate the comment. We previously published a study that compared the diagnosis performance of lung ultrasound versus ReDS to find pulmonary congestion (Heart Vessels 2022. PMID: 36258045). In this previous study, ReDS values and lung ultrasound had a weak correlation. Nevertheless, both modalities showed stronger correlation in patients with severe pulmonary congestion. We revised the discussion section.
Before 4
4.1 ReDS values and CSIs to evaluate pulmonary congestion:
ReDS values have a moderate correlation with high-resolution computed tomography and a mild correlation with invasively measured pulmonary artery wedge pressure.
After 4
4.1 ReDS values and CSIs to evaluate pulmonary congestion:
ReDS values have a moderate correlation with high-resolution computed tomography, a mild correlation with invasively measured pulmonary artery wedge pressure, and weak correlations with plasma B-type natriuretic peptide levels and lung ultrasound [10-12].
Comment 5
Clinical signs and symptoms of congestion (i.e., leg edema, jugular venous distention, and New York Heart Association class) were lacking in the current analysis.
Response 5
Thank you for your comment. Given that not all included patients had heart failure, we did not routinely obtain heart failure signs. We do not deny at all the clinical implication to obtain physical signs to estimate pulmonary congestion, but pulmonary congestion is often challenging to be accurately estimated by physical examination alone (Narang N, et al. J Card Fail. 2020;26(2):128-135). Several physical signs including peripheral edema and jugular venous distension might be particularly useful to assess “peripheral congestion”, rather than “pulmonary congestion”. In this study, we focused on the assessment of “pulmonary congestion”, and physical examination might not necessarily be useful to assess it.
Anyway, we do not have physical examination data in all participants, which should be one of the critical limitations. We added the following sentence in the limitation section.
Before 5
4.3 Study limitation
None.
After 5
4.3 Study limitation
We did not routinely obtain physical examinations to assess pulmonary congestion, although it is sometimes challenging to accurately estimate the degree of pulmonary congestion by physical examinations alone.
Comment 6
>70% of patients were not diagnosed with HF, but had certain degrees of ReDS-derived pulmonary congestion as presented in Table 1. So, what is a diagnosis for such patients? In general, patients who have symptomatic congestion are defined as HF.
Response 6
We appreciate your comment. We believe that ReDS technology can distinguish a variety of severity of pulmonary congestion, ranging between sub-clinical congestion and symptomatic and severe congestion. We strengthened the data of comorbidities in Table 1. Many patients had a variety of comorbidities other than heart failure, including valvular diseases, CKD, and atrial fibrillation. These comorbidities would also be associated with incremental lung fluid levels, although most of them might be asymptomatic.
Before 6
See previous Table 1
4.2 Clinical implication of ReDS measurement:
None
4.2 Clinical implication of ReDS measurement:
None
After 6
Table 1. Baseline characteristics
|
|
All (N = 458) |
Non-heart failure (N = 328) |
Heart failure (N = 130) |
|
Past medical history |
|
|
|
|
・・・・ |
・・・ |
・・・ |
・・・ |
|
Diabetes mellitus |
163 (36%) |
111 (34%) |
52 (40%) |
|
Valvular diseases |
149 (33%) |
90 (27%) |
59 (45%) |
|
Chronic kidney diseases |
305 (66%) |
200 (61%) |
105 (81%) |
|
Atrial fibrillation |
161 (35%) |
102 (31%) |
59 (45%) |
4.2 Clinical implication of ReDS measurement:
Such a mild pulmonary congestion might also be caused by non-HF etiologies including chronic kidney diseases.
4.2 Clinical implication of ReDS measurement:
Nevertheless, there are no studies that directly investigated the clinical implication of mild pulmonary congestion assessed by ReDS.
Comment 7
Renal function may influence the relationship between ReDS and CSI particularly in patients with HF.
Response 7
We appreciate your comment. We added the prevalence of CKD in Table 1. The existence of CKD contributes to systemic congestion. Thus, the degree of pulmonary congestion might be indirectly and partially affected by the existence of CKD.
Before 7
See previous Table 1
After 7
See updated Table 1
Comment 8
Please check prevalence rate of prior heart failure in Table 1. It may be typo error.
Response 8
Thank you for your comment. Since the total number of patients is 458 cases with 130 heart failure patients, it seemed to us that 28% is correct.
Before 8
None.
After 8
None.

This manuscript is a resubmission of an earlier submission. The following is a list of the peer review reports and author responses from that submission.
Round 1
Reviewer 1 Report
Dr. Toshihide Izumida presented in this paper the data which addressed a very important item, evaluation of the accuracy of lung pulmonary congestion by new technology ReDS.
Authors collected data from 458 patients admitted to the hospital. 130 of them had a history of HF and 328 patients had no history of HF. CXR score is assessed in 6 different lung fields. The method was published previously and was found suitable for quantification of lung pulmonary congestion.
Table 1 and Figure 3 are the only sources of the data.
Serious questions:
Under Table 1 is written:” Continuous variables were expressed as median and interquartile or mean ± standard deviation 104 and categorical variables were expressed as numbers and percentages”. I didn’t find the data which is presented as mean ± standard deviation.
Presentation data as median and interquartile makes checking the data impossible. Anyway, there are some serious remarks.
1. ReDS index = 28% is absolutely identical between “ALL” patients (N=458), non-heart failure patients (N=328) and heart failure patients (N=130). First question is: if the dada is right, how does the ReDS index differentiate between non HF and HF patients. Important that the CXR score between groups is different!
2. Second question is: in the text of the paper there is an additional subgroup of N=67 patients with significant pulmonary congestion, which is the part of HF patients group (N=130). There is no data about this subgroup at all, but the most important conclusion is done exactly about this subgroup.
3. Third question. The ReDS index in the HF group is 28 (25,33)%. Normal ReDS index corresponding to the normal lung fluid status is 20-35%. Question is, were patients of HF group congested? The CXR score for this group is 0.08(0.00, 0.27). How do we interpret this value?
4. Authors conclude that “the ReDS technology might be a practical alternative tool to non-invasively quantify pulmonary congestion without exert technique, particularly for the HF cohorts” (177). Data presented in this paper support the opposite conclusion, that the ReDS index is equal between HF and non-HF groups.
Small points:
1. Frequencies of dyslipidemia in a group of “all” patients = 55%. In non HF patients = 34% and in HF group = 53%. Mathematically it's impossible. The same mistake is about the congestion score index.
Author Response
Reviewer-1.
Dr. Toshihide Izumida presented in this paper the data which addressed a very important item, evaluation of the accuracy of lung pulmonary congestion by new technology ReDS.
Authors collected data from 458 patients admitted to the hospital. 130 of them had a history of HF and 328 patients had no history of HF. CXR score is assessed in 6 different lung fields. The method was published previously and was found suitable for quantification of lung pulmonary congestion.
Table 1 and Figure 3 are the only sources of the data.
Response.
We express our great appreciation for the reviewer’s critical suggestions and comments. We attempted our best to answer all the comments and revise our draft. We also corrected typographs and grammatical errors. We believe that the revised draft would be suitable for the publication in Journal of Clinical Medicine.
Comment1.
Serious questions:
Under Table 1 is written:” Continuous variables were expressed as median and interquartile or mean ± standard deviation 104 and categorical variables were expressed as numbers and percentages”. I didn’t find the data which is presented as mean ± standard deviation.
Presentation data as median and interquartile makes checking the data impossible. Anyway, there are some serious remarks.
Response1.
Thank you for your comment. We corrected mistakes in the table and figure. Since our data was small sample, we used median to present all continuous variables irrespective of their distributions.
Before1.
Table 1.
Continuous variables were expressed as median and interquartile or mean ± standard deviation.
Statistical analysis:
Continuous variables were presented as median and interquartile for non-parametric distributions and mean ± standard deviation for parametric distributions.
After1.
Table1.
Continuous variables were expressed as median and interquartile.
Statistical analysis:
Continuous variables were presented as median and interquartile.
Comment2.
ReDS index = 28% is absolutely identical between “ALL” patients (N=458), non-heart failure patients (N=328) and heart failure patients (N=130). First question is: if the dada is right, how does the ReDS index differentiate between non HF and HF patients. Important that the CXR score between groups is different!
Response2.
Thanks for your comment. First of all, the essence of our study is to investigate the correlation between the CSI of chest X rays and ReDS to assess the degree of pulmonary congetison, not to compare the HF cohort with non-HF cohort. ReDS is not a system to find HF patients, but to evaluate just the pulmonary congestion. Since both stable heart failure and non-heart failure patients generally have almost no pulmonary congestion, the lack of significant differences in ReDS values between the two groups is not so surprising.
Before2.
None
After2.
Discussion.
4.1 ReDS values and CSIs to evaluate pulmonary congestion
Since both stable heart failure and non-heart failure patients generally have almost no pulmonary congestion, ReDS cannot distinguish between them.
Comment3.
Second question is: in the text of the paper there is an additional subgroup of N=67 patients with significant pulmonary congestion, which is the part of HF patients group (N=130). There is no data about this subgroup at all, but the most important conclusion is done exactly about this subgroup.
Response3.
Thank you for your comment. The essence of our study is to investigate the association between the CSI of chest X ray and ReDS to detect pulmonary congetison, not to compare patients with congestion, with patients without congestion. The subgroups were classified not to examine characteristics between groups, but to compare and evaluate pulmonary congestion in ReDS and chest X-rays in each cohort and investigate the robustness of the results. As you pointed out, for one of the main results, a supplementary table was added.
Before3.
None.
After3.
Appendix
Table A1. Baseline data
|
|
Patients with pulmonary congestion (N = 67) |
Patients without pulmonary congestion (N =391) |
|
Demographics |
|
|
|
Age, years |
73 (64, 83) |
76 (69, 82) |
|
Man |
39 (58%) |
228 (58%) |
|
Height, cm |
159 (150, 167) |
159 (151, 166) |
|
Body mass index, kg/m2 |
23.8 (21.9, 27.8) |
22.5 (20.2, 24.8) |
|
Laboratory data |
|
|
|
Hemoglobin, g/dL |
12.4 (11.0, 13.4) |
12.5 (11.2, 13.8) |
|
Serum albumin, g/dL |
3.8 (3.4, 4.1) |
3.9 (3.6, 4.2) |
|
Serum creatinine, mg/dL |
0.9 (0.8, 1.5) |
1.0 (0.8, 1.5) |
|
Plasma B-type natriuretic peptide, pg/mL |
124 (63, 405) |
89 (30, 233) |
|
Echocardiographic data |
|
|
|
Left ventricular ejection fraction, % |
60 (49, 69) |
62 (51, 69) |
|
Left ventricular end-diastolic diameter, mm |
49 (45, 56) |
48 (43, 52) |
|
Left ventricular end-systolic diameter, mm |
33 (28, 42) |
31 (27, 37) |
|
Left atrial diameter, mm |
44 (38, 52) |
40 (34, 46) |
|
Past medical history |
|
|
|
Heart failure |
23 (34%) |
107 (27%) |
|
Stroke |
14 (21%) |
70 (18%) |
|
History of coronary intervention |
17 (25%) |
90 (23%) |
|
Hypertension |
47 (70%) |
292 (75%) |
|
Dyslipidemia |
35 (52%) |
216 (55%) |
|
Diabetes mellitus |
27 (40%) |
136 (35%) |
|
ReDS, % |
37 (36, 39) |
27 (24, 30) |
|
Chest X-ray |
|
|
|
Congestion score index |
0.17 (0.00, 0.73) |
0.00 (0.00, 0.17) |
Continuous variables were expressed as median and interquartile and categorical variables were expressed as numbers and percentages.
Comment4.
Third question. The ReDS index in the HF group is 28 (25,33)%. Normal ReDS index corresponding to the normal lung fluid status is 20-35%. Question is, were patients of HF group congested? The CXR score for this group is 0.08(0.00, 0.27). How do we interpret this value?
Response4.
Thanks for your comment. Since stable heart failure patients generally have almost no or only a slight degree of pulmonary congestion, ReDS values would be within normal in many heart failure patients. The CSI was 0.08 (0.00, 0.27) in this cohort. This value would be also clinically trivial.
Before4.
None.
After4.
4.1 ReDS values and CSIs to evaluate pulmonary congestion
Since both stable heart failure and non-heart failure patients generally have almost no pulmonary congestion, ReDS cannot distinguish between them.
Comment5.
Authors conclude that “the ReDS technology might be a practical alternative tool to non-invasively quantify pulmonary congestion without exert technique, particularly for the HF cohorts” (177). Data presented in this paper support the opposite conclusion, that the ReDS index is equal between HF and non-HF groups.
Response5.
Thank you for your comment. We corrected the conclusion section.
Before5.
Conclusions
The ReDS technology might be a practical alternative tool to non-invasively quantify pulmonary congestion without exert technique, particularly for the HF cohorts.
After5.
Conclusions
The ReDS technology might be a practical alternative tool to non-invasively quantify pulmonary congestion without exert technique.
Comment6.
Small points. Frequencies of dyslipidemia in a group of “all” patients = 55%. In non HF patients = 34% and in HF group = 53%. Mathematically it's impossible. The same mistake is about the congestion score index.
Response6.
We corrected the mistake. However, only the frequency of dyslipidemia was wrong, the other items were fine. The congestion score index is a non-parametric distribution and was not wrong. A histogram is also attached.
Before6.
Table 1
Frequencies of dyslipidemia in a group of “all” patients = 55%. In non HF patients = 34% and in HF group = 53%.
After6.
Table 1
Frequencies of dyslipidemia in a group of “all” patients = 55%. In non HF patients = 55% and in HF group = 53%.

Reviewer 2 Report
This is interesting article about remote dielectric sensing (ReDS) which is a recently-introduced non-invasive electromagnetic energy-based technology to quantify pulmonary congestion. Authors correlate ReDS and chest X-ray examination in the 458 patients with pulmonary congestion.They conclude that the ReDS technology might be a practical tool to non-invasively quantify pulmonary congestion.
Author Response
Reviewer-2.
This is interesting article about remote dielectric sensing (ReDS) which is a recently-introduced non-invasive electromagnetic energy-based technology to quantify pulmonary congestion. Authors correlate ReDS and chest X-ray examination in the 458 patients with pulmonary congestion.They conclude that the ReDS technology might be a practical tool to non-invasively quantify pulmonary congestion.
Response.
We express our great appreciation. We corrected typographs and grammatical errors. We believe that the revised draft would be suitable for the publication in Journal of Clinical Medicine.
